## Research Article

HIV; suicide screening; Tanzania; suicide prevention; measures validation

**Corresponding author:**
Brandon A. Knettel;
Email: Brandon.Knettel@duke.edu

# Validation of a culturally sensitive, Swahili-translated instrument to assess suicide risk among adults living with HIV in Tanzania

Linda Minja[1], Brandon A. Knettel[2,3] ⓘ, Wei Pan[2,4], Kim Madundo[5,6], Ismail Amiri[1,3,5], Louise Joel[5], Elizabeth Knippler[2,7], Michael V. Relf[2,3], Joao Ricardo N. Vissoci[3,8], Catherine A. Staton[3,8], Elizabeth F. Msoka[1,5,6], Clotilda S. Tarimo[1], Victor Katiti[1,6], Blandina T. Mmbaga[1,3,5,6] and David B. Goldston[9]

[1]Kilimanjaro Clinical Research Institute, Moshi, Tanzania; [2]Duke University School of Nursing, Durham, NC, USA; [3]Duke Global Health Institute, Duke University, Durham, NC, USA; [4]Department of Population Health Sciences, Duke University, Durham, NC, USA; [5]Kilimanjaro Christian Medical Centre, Moshi, Tanzania; [6]Kilimanjaro Christian Medical University College, Moshi, Tanzania; [7]Duke Center for AIDS Research, Duke University, Durham, NC, USA; [8]Duke Department of Emergency Medicine, Durham, NC, USA and [9]Department of Psychiatry & Behavioral Sciences, Duke University, Durham, NC, USA

## Abstract

In Tanzania, there are high rates of suicidal thoughts and behavior among people living with HIV (PLWH), yet few instruments exist for effective screening and referral. To address this gap, we developed and validated Swahili translations of the Columbia Suicide Severity Rating Scale (C-SSRS) Screen Version and two accompanying scales assessing self-efficacy to avoid suicidal action and reasons for living. We administered a structured survey to 80 PLWH attending two HIV clinics in Moshi, Tanzania. Factor analysis of the items revealed four subscales: suicide intensity, self-efficacy to avoid suicide, fear and social concern about suicide, and family and spirituality deterrents to suicide. The area under the receiver operating curve showed only suicide intensity, and fear and social concern met the prespecified cutoff of ≥0.7 in accurately identifying patients with a plan and intent to act on suicidal thoughts. This study provides early evidence that brief screening of intensity of suicidality in the past month, assessed by the C-SSRS Screen Version, is a strong, resource-efficient strategy for identifying suicide risk in the Tanzanian setting. Patients who report little fear of dying and low concern about social perceptions of suicide may also be at increased risk.

## Impact statement

Suicide is a major cause of loss of life worldwide, claiming about 700,000 lives every year. Many African nations have high suicide rates and few mental health resources, leading to low levels of suicide screening and poor linkage to care. In Tanzania, there is fewer than one psychiatrist or psychologist per one million people. Given these human resource constraints, the "gold standard" of a full clinical assessment by a mental health professional is rarely feasible. Furthermore, there is a lack of locally validated tools to screen for suicidal thinking and behavior. In this study, we explored whether adapted brief measures of suicidal intensity, self-efficacy to avoid suicide, and reasons for living could be used to effectively assess suicide risk in Tanzanian HIV clinics. Brief assessment tools can be administered by nonspecialists in a variety of settings, which can greatly enhance the recognition of people at risk for suicide and facilitate linkage to care. We collected data from 80 people living with HIV, a group that has a remarkably high prevalence of suicide compared with the general population. Participants were screened for suicidal ideation during routine HIV care appointments, and those who reported suicidal ideation in the past month were enrolled and completed a structured survey. As expected, a higher intensity of suicidality was strongly associated with suicide risk (i.e., active plan or intent to attempt suicide). Items from the reasons for living measure that assessed fear of death and concerns about social reactions to suicide were also associated with risk. These brief suicide measures are highly feasible for administration by nonspecialists and show promise for identifying the risk of suicide. Implementing routine suicide risk screening in HIV care and other high-risk settings is a low-impact strategy to rapidly improve the assessment of suicide, enhance linkage to mental health care, and save lives.

## Background

Suicide is a major cause of premature death, as more than 700,000 people die by suicide each year worldwide (World Health Organization, 2021). More than three-fourths of these deaths occur in

low- and middle-income countries (LMICs) (Bachani et al., 2020). Rates of suicide in sub-Saharan Africa (SSA) are among the highest in the world (Naghavi, 2019; United Nations, 2022), and the topic of suicide has been increasingly prominent in news reports and public discourse in SSA in the past decade (Makoye, 2021; Muiruri, 2021). However, little attention has been paid to improving assessment and evidence-based intervention for suicide prevention in the region, and the evidence obtained from countries outside of Africa may not be culturally appropriate (Mars et al., 2014; Knettel et al., 2023c).

Public accounts of suicide in SSA often focus on the events leading up to the death, such as stress due to relationship conflicts, financial challenges, physical illness, substance use, and unattended mental health issues (Mars et al., 2014; Li et al., 2021; Kaggwa, 2022). Chronic illness is another major contributor to suicide risk globally, particularly in low-resource settings where preventable deaths are all too common (Bachmann, 2018). SSA is the region of the world with the greatest burden of HIV, and death by suicide is shockingly common among people living with HIV (PLWH), representing a true public health emergency (Ndosi et al., 2004; Pelton et al., 2021).

Suicidal behavior, including attempting suicide, is illegal in nine African countries, including Tanzania (Mishara & Weisstub, 2015). The risk of legal consequences surrounding suicide contributes to widespread societal stigma around this issue and may lead to considerable underreporting of suicidal thoughts, behaviors, and deaths (Naghavi, 2019; Uddin et al., 2019). This also means that many people experiencing suicidal thoughts do not access mental health services due to a credible fear of prosecution (Moradinazar et al., 2019).

In Tanzania, only 55 psychiatrists and psychologists attempt to provide care for a population of more than 60 million people (World Health Organization, 2020). Extremely high rates of suicidal thinking have been identified in HIV care in Tanzania (Knettel et al., 2020), yet very few resources are available for PLWH who are experiencing a mental health crisis (Oshosen et al., 2020; Madundo et al., 2023). The few mental providers in the country are mostly based at large, urban tertiary facilities, which are costly and often difficult to access (Binyaruka & Borghi, 2022). Furthermore, few models exist for effective suicide intervention in SSA, and even fewer address the unique experiences of PLWH (Govender et al., 2014; Knettel et al., 2023c).

In the context of these multiple barriers to appropriate care for people experiencing suicidality, there is an urgent need for an improved, culturally sensitive assessment of suicide risk and linkage to treatment. There is also a notable lack of locally validated assessment tools for suicide risk in SSA. Several studies have relied on a single item related to suicidal thoughts, such as the final question of the Patient Health Questionnaire-9 (Doukani et al., 2021; Hammett et al., 2022), or measures that do not measure suicidal intent, such as Module C of the Mini-International Neuropsychiatric Interview (Ertl et al., 2011). Others have used longer measures such as the Beck Scale for Suicide Ideation or the Suicide Risk Screening Scale (Govender et al., 2014; Mutiso et al., 2019) that take more time and effort to administer, which can present challenges in a busy clinical setting. Further, few studies in Africa have utilized strengths-focused suicide assessments such as measures of self-efficacy to avoid suicide or reasons for living, which include culturally salient constructs in African settings such as the influence of religious beliefs and collectivistic social values on suicide risk (Lawrence et al., 2016; Pompili, 2022).

The objective of this study was to validate a brief, culturally informed, Swahili-translated measure of suicide risk, the Columbia Suicide Severity Rating Scale (C-SSRS) Screen Version (Posner et al., 2011; Andreotti et al., 2020) among adults living with HIV in Tanzania. To capture a broader view of suicide risk, we also examined the value of adding items assessing reasons for living and self-efficacy to avoid suicide. We hypothesized that the combined measure would provide a valid and reliable brief screen for suicide risk among PLWH that could be administered by healthcare workers in Tanzania.

## Methods

We conducted a cross-sectional survey study with 80 adults who were living with HIV and experiencing suicidal ideation at two urban HIV clinics in Moshi, Tanzania, from January to October of 2022. These data were collected as part of a larger study aiming to develop a task-shifted counseling intervention focused on preventing suicide and improving HIV care engagement in Tanzania (Knettel et al., 2023b).

The two study clinics provide routine HIV care for approximately 6,000 PLWH within the region according to national protocols, whereby all PLWH receive HIV care and medication free of charge. For patients experiencing emotional distress, brief counseling is provided by professional nurses during HIV care, and people with severe distress are referred for psychiatric care. However, before the commencement of this study, no routine suicide risk assessment was occurring in these clinics.

## Procedures

At the start of the study, we provided training to the HIV clinic staff at these two sites to implement suicide screening during all routine HIV clinic appointments. Professional nurses administered a single yes/no screening item derived from the C-SSRS Screen Version (Posner et al., 2011): "In the last month, have you had any actual thoughts of killing yourself?" Patients who responded yes to the screening item were informed of the research and, if interested, were referred to meet with a study research assistant (RA). Participants were eligible for inclusion if they were 18 years of age or older, fluent in Swahili or English, and deemed medically and cognitively capable of completing the study procedures according to the HIV clinic nurses and study RA.

Upon referral from the clinic nurses, the RA read the consent form aloud and interested, eligible patients provided written informed consent before enrollment. Patients who responded yes to the screening item but were not interested in participating or were ineligible for participation were referred for psychiatric treatment according to the standard of care.

Upon enrollment, the RA verbally administered a structured, tablet-based survey to the participant. Participant responses were entered in real time into a secure cloud-based data repository. The C-SSRS component of the survey was audio-recorded, and select recordings were reviewed for quality assurance.

Upon completion of the study procedures, the RA provided an approximately 20-minute safety planning session using a structured worksheet according to evidence-based procedures developed by Stanley and Brown (2012). All participants then received referral information for standard-of-care psychiatry services. The research team had procedures in place for further support of any individuals with an active plan or intent to attempt suicide at the

conclusion of their participation; however, this was not needed for any of our study participants.

Study procedures were approved by the Tanzanian National Institute for Medical Research and the ethical review boards of Kilimanjaro Christian Medical Centre and the Duke University Health System.

### Research assistant characteristics and training

The study RAs included two individuals with bachelor's level training in psychology and prior training and experience in psychology and one study nurse. RAs received two weeks of study-specific training from the principal investigator, who is a licensed psychologist, and a consultant psychiatrist before any patient contact. Training included didactic training in mental health and HIV, suicidality, safety planning intervention, counseling skills, suicide assessment, and steps to mitigate and respond to suicide risk in research. RAs completed several mock assessments during training and were required to demonstrate competency in these mock sessions before enrolling actual study participants.

The RAs also attended a weekly supervision session with a psychologist and two psychiatrists (study supervisors) for quality assurance and skill building. Each week, 1 to 2 audio recordings of C-SSRS assessments were reviewed by the three study supervisors using an adapted Therapy Quality Scale (Patel et al., 2017) to evaluate the quality of the assessments and provide structured feedback.

### Measures

All measures were translated from English to Swahili by a bilingual research team member and then back-translated by a second team member. The full study team then cross-checked the versions and made edits for clarity and cultural appropriateness until team consensus was reached.

#### Sociodemographic variables

We first collected a variety of sociodemographic variables, including the participant's age, gender, religion, education, relationship status, employment status, and monthly income.

#### Intensity of suicidality

The intensity of recent suicidality was assessed using the Columbia Suicide Severity Rating Scale (C-SSRS) Screen Version and Full Version (Posner et al., 2011). The Screen Version has six yes/no items that use plain language to assess suicidal thinking, intent, and preparatory behavior (e.g., writing a suicide note or gathering materials needed to attempt suicide) in the past one month. Items include "Have you had suicidal thoughts and had some intention of acting on them?" and "Have you started to work out or worked out the details of how to kill yourself? Do you intend to carry out this plan?" Cronbach's alpha for the C-SSRS Screen Version items in this study was 0.84.

The C-SSRS Full Version is a semi-structured guide for a longer, more narrative clinical interview, intended to support the gathering of details about the intensity of suicidal thinking, suicide risk, any past suicide attempts or self-injurious behaviors, and lethality of those attempts. The last item in the Full Version is a rating by the interviewer of the patient's highest severity of suicidality in the past

month, which serves as a summary risk assessment. The rating on this item (C-SSRS_M) ranges from 1 (wish to be dead) to 9 (actual attempt). Information obtained from the Full Version, particularly the determination from the clinical interview of whether a participant had an active plan or intent to attempt suicide, was used as a gold standard for suicide risk assessment in this study.

#### Self-efficacy to avoid suicide

Participant belief in his/her ability to avoid acting upon suicidal thoughts was assessed by the Self-Efficacy to Avoid Suicidal Action (SEASA) scale. This scale consists of six questions (e.g., "How certain are you that you could control future thoughts of suicide if you were experiencing physical or emotional pain?") that are rated from 0 (very uncertain) to 10 (very certain), with higher scores indicating higher self-efficacy. Cronbach's alpha for the SEASA items in this study was 0.91.

#### Reasons for living

Motivations to stay alive and not to attempt suicide were measured by the Brief Reasons for Living (BRFL) inventory, which consists of twelve questions (e.g., "My family depends upon me and needs me"), with response options starting from 1 (not at all important) to 6 (very important) and higher scores indicating higher endorsement of each reason for living. Cronbach's alpha for the BRFL items in this study was 0.79.

### Statistical analysis

We summarized sociodemographic characteristics using means and standard deviations or frequencies and percentages as appropriate (Table 1). We conducted an exploratory factor analysis (EFA) to assess validity related to the internal structure of the combined items from the C-SSRS Screener, SEASA, and BRFL. Our sample size was fairly small. However, common standards for acceptable sample size in EFA are a minimum of 50 participants or three participants per item (de Winter et al., 2009), both of which were met with our sample of 80 participants for 22 items (Tables 2 and 3).

Because the scales had different response options, we created a polychoric correlation matrix among the scales for data input. Polychoric correlations are preferred over Pearson's correlations when dealing with ordinal variables or a combination of ordinal, binary, and continuous ones (Holgado–Tello et al., 2010; Garcia-Santillan et al., 2021). In our case, as we had both ordinal and binary variables, we chose to utilize a polychoric correlation matrix. Items 1 and 2 from the C-SSRS screening version (related to recent thoughts of suicide) were part of the enrollment criteria and relevant to all participants, so these were not included in the analysis. The resulting polychoric correlation matrix was non-positive definite, and therefore, smoothing was performed (see Supplementary materials).

Parallel analysis and scree plot were used to suggest the number of factors to extract and factor analysis was then conducted using oblique (oblimin) rotation, assuming that the subscales are correlated. Items were retained if they had a loading of >0.35, a communality of ≥0.25, and no cross-loading. Cross-loading was defined as loading >0.35 in more than one factor and a ratio of the square of the loadings (variance) below 2.0 (Hair & Babin, 2018).

**Table 1.** Sociodemographic characteristics and HIV history (*N*=80)

| Characteristics | *n* (%) |
|---|---|
| Age [median (Q1, Q3)] | 42 (36, 49) |
| Gender | |
| Female | 62 (77.5) |
| Male | 18 (22.5) |
| Religion | |
| Christian | 58 (72.5) |
| Muslim | 22 (27.5) |
| Education | |
| No formal education | 3 (3.8) |
| Primary | 58 (72.5) |
| Secondary | 16 (20.0) |
| Post-secondary | 3 (3.8) |
| Relationships status | |
| Married/cohabiting | 34 (42.5) |
| Single/divorced/widow/separated | 46 (57.5) |
| Are you currently working? | |
| No | 29 (36.2) |
| Yes | 51 (63.7) |
| Monthly income [median (Q1, Q3)] | 60,000 Tsh (30,000, 200,000) |
| Were you born with HIV? | |
| No | 76 (95.0) |
| Yes | 3 (3.8) |
| Don't know | 1 (1.3) |
| Have you ever told another person about your HIV status? | |
| No | 13 (16.2) |
| Yes | 67 (83.8) |

To examine criterion validity, adjusted $R^2$ was obtained from a regression model with the extracted factors as predictors and the final summary item of risk (C-SSRS_M) as the dependent variable. We hypothesized that there would be significant associations between factors derived from all three included scales (C-SSRS Screen Version, SEASA, and BRFL) with the C-SSRS_M summary item of suicide risk. Additionally, to evaluate the scale's capability to identify patients who are at high risk, we calculated the area under receiver operating characteristic (AUROC) curves (see Supplementary materials). An AUROC ≥ .70 indicates that the scale will produce acceptable discrimination as a diagnostic test (Hosmer et al., 2013).

## Results

Eighty PLWH were enrolled in this study. The majority were women (n=62, 77.5%), and the median age of this sample was 42 years. Most participants had a primary school education or less (n=61, 76.3%) and were not in a relationship (i.e., were single,

divorced, or widowed) (n=46, 57.5%). The median monthly income was 60,000 Tanzanian shillings (Tsh), equivalent to roughly $26 USD per month. Disclosure of one's HIV status was common in this sample, as 83.8% (n=67) of participants had told at least one other person that they were living with HIV.

The scree plot suggested that the best fit would occur with four factors, and parallel analysis suggested that the best fit would occur with three to five factors; all three solutions were examined (see Supplementary Materials). The variance explained by the solutions was 62%, 62%, and 67%, for the 3-factor, 4-factor, and 5-factor solutions, respectively. The 4-factor solution was determined to be most adequate, as all of the original scale items were retained in this solution, whereas theoretically important items had low loadings in the 3- and 5-factor solutions (Table 2).

In the final 4-factor solution, we retained one item with a communality below 0.25, "If you have thoughts of killing yourself in the future, how confident are you that you will tell someone?", as this item measures a very important aspect of self-efficacy to avoid suicidal action. It was determined that the low communality for this item was likely related to the difficulty of disclosing suicidality, particularly in a culture where social ties are critically important to emotional health and suicidal thinking is both highly stigmatized and illegal.

In the final 4-factor solution, items from the Self-Efficacy to Avoid Suicidal Action scale formed Factor 1, "Self-Efficacy" (α = 0.98), and items from the C-SSRS screening version formed Factor 2, "Intensity of Suicidality" (α = 0.99). Items from the Brief Reasons for Living inventory were split, with items related to one's family, children, and spirituality making up Factor 3, "Family and Spirituality" (α = 0.94), and items related to fear, morality, and social perception making up Factor 4, "Fear and Social Concern" (α = 0.92). Cronbach's alpha for the combined 4-factor scale was 0.91 and the corrected item–total correlation for all items was above 0.3, indicating an adequate correlation between items and the overall scale, despite the items deriving from three different instruments.

Inter-factor correlations between "Intensity of Suicidality" subscale and other subscales were -0.42 for "Self-Efficacy," -0.17 for "Family and Spirituality," and -0.25 for "Fear and Social Concern." All inter-factor correlations were in theoretically expected directions (Table 3).

As expected, regression analysis showed that a higher Intensity of Suicidality (derived from the C-SSRS Screen Version) was strongly associated with suicide risk (i.e., active plan or intent to attempt suicide, derived from the C-SSRS Full Version) (β=0.91; 95% CI= 0.69, 1.13). The Fear and Social Concern subscale (β=-0.07; 95% CI= -0.13, -0.01) was significantly negatively correlated with suicide risk. However, Self-Efficacy (β=0.003; 95% CI= -2.89, 3.92) and Family and Spirituality (β=0.04; 95% CI= -0.06, 1.14) were not significantly associated with suicide risk (Figure 1). The total variance in suicide risk explained by the four factors was 55.9% and 54.5% when Intensity of Suicidality was the only predictor. Both the Akaike information criterion (AIC) and Bayesian information criterion (BIC) indicated that Intensity of Suicidality alone is the best model for identifying suicide risk.

Similarly, ROC analysis showed that Intensity of Suicidality was most useful in correctly identifying patients with high suicide risk [AUC = 0.89], followed by Fear and Social Concern [AUC = 0.78], Self-Efficacy [AUC = 0.69], and lastly Family and Spirituality [AUC = 0.64].

To utilize the individual subscales as a screening test, optimal cutoff points of the significant predictors can be referenced to identify individuals at high risk of suicide and the need for

**Table 2.** Exploratory factor analysis of the combined items, final 4-factor solution

| | | Factor 1 | Factor 2 | Factor 3 | Factor 4 |
|---|---|---|---|---|---|
| seasa2 | If you have serious thoughts of killing yourself in the future, how confident are you that you will be able to keep yourself from attempting suicide? | 0.94 | | | |
| seasa4 | How certain are you that you could control future thoughts of suicide if you were experiencing physical or emotional pain? | 0.89 | | | |
| seasa1 | How confident are you that you will not attempt suicide in the future? | 0.87 | | | |
| seasa5 | How certain are you that you could control future suicidal thoughts if you lost an important relationship? | 0.84 | | | |
| seasa6 | How certain are you that you could control future suicidal thoughts if you lost a job, could not find employment, or suffered a financial crisis? | 0.69 | | | |
| seasa3 | If you have thoughts of killing yourself in the future, how confident are you that you will tell someone? | 0.42 | | | |
| cssrs5 | Have you started to work out or worked out the details of how to kill yourself? Do you intend to carry out this plan? | | 0.94 | | |
| cssrs6 | Have you ever done anything, started to do anything, or prepared to do anything to end your life? | | 0.89 | | |
| cssrs4 | Have you had these thoughts and had some intention of acting on them? | | 0.84 | | |
| cssrs3 | Have you been thinking about how you might do this? | | 0.77 | | |
| brfl7 | I want to watch my children as they grow | | | 0.82 | |
| brfl5 | I love and enjoy my family too much and could not leave them | | | 0.81 | |
| brfl4 | The effect on my children could be harmful | | | 0.77 | |
| brfl2 | My family depends upon me and needs me | | | 0.76 | |
| brfl12 | I believe I can find purpose in life, a reason to live | | | 0.43 | |
| brfl6 | My religious beliefs forbid it | | | 0.36 | |
| brfl8 | I am concerned about what others would think of me | | | | 0.71 |
| brfl10 | I am afraid of the actual "act" of killing myself (the pain, blood, violence) | | | | 0.67 |
| brfl11 | I would not want people to think I did not have control over my life | | | | 0.64 |
| brfl9 | I consider it morally wrong | | | | 0.35 |
| brfl1 | I am afraid of death | | | | 0.49 |
| brfl3 | I do not want to die | | | | 0.42 |

**Table 3.** Inter-factor correlations

| | Intensity of suicidality | Family and spirituality | Fear and social concern |
|---|---|---|---|
| Family and spirituality | −0.17 | | |
| Fear and social concern | −0.25 | 0.26 | |
| Self-efficacy | −0.42 | 0.39 | 0.24 |

intervention. For Intensity of Suicidality, our findings support the commonly accepted cut point used for the C-SSRS Screen Version, which is a "yes" response on any of items 3, 4, 5, or 6, indicating active plan, intent, or preparation for a suicide attempt. For the Fear and Social Concern subscale, a score ≤ 34 may indicate a higher risk of attempting suicide.

## Discussion

Constraints in healthcare worker capacity and a lack of specialist mental health providers are major barriers to suicide risk assessment in many low- and middle-income countries. In Tanzania, we have identified high rates of suicidality among PLWH, but mental health providers are extremely rare, HIV clinic staff have little mental health training, and healthcare workers report feeling overburdened in their current roles (Knettel et al., 2018; Oshosen et al., 2020). In light of these challenges, the "gold standard" of an extended clinical interview by a mental health professional to assess suicide risk is rarely feasible (Knettel et al., 2023b). In this analysis, we identified that the brief screening of Intensity of Suicidality in the past month, measured by the C-SSRS Screen Version, is a valid and reliable strategy for identifying suicide risk (plan or intent to attempt suicide) among PLWH in Tanzania.

In addition to measures of recent suicidal intensity, strengths-focused instruments such as the BRFL can be beneficial in assessing suicide risk (Bakhiyi et al., 2016; Malone et al., 2000), as these are more likely to capture culturally salient protective factors (Lawrence et al., 2016; Pompili, 2022). In our analysis, participants with low endorsement of fear of death and low social concern on the BRFL were more likely to have an active plan or intent to act on suicidal thoughts. Prior studies have shown that reduced fear of death differentiates between having suicide ideation and attempting suicide (Dhingra et al., 2015; Smith et al.,

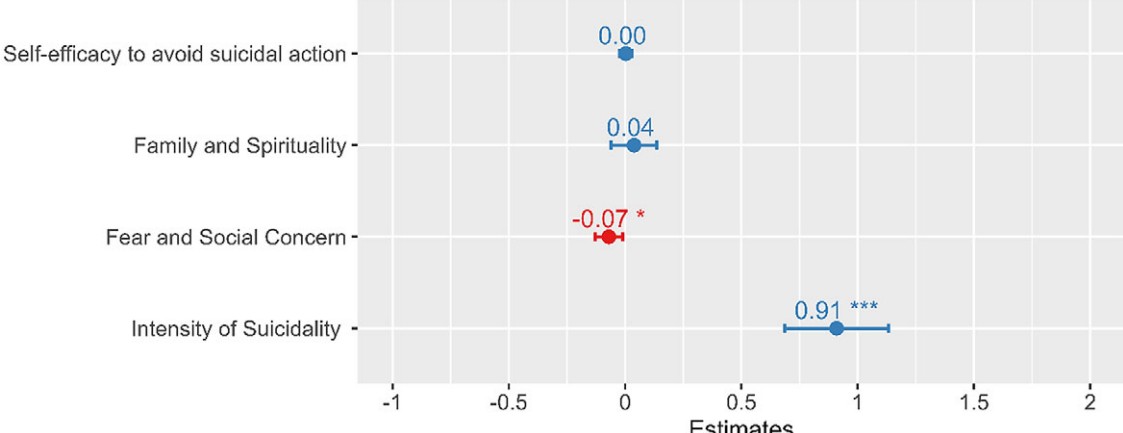

**Figure 1.** Suicide risk severity prediction by the four subscales.

2016; Klonsky et al., 2017). This highlights the potential of these questions in identifying patients who need immediate help and referral for a higher level of care. The Family and Spirituality subscale of the BRFL was not associated with reduced suicide risk in the current study. However, understanding how the patient views family and spirituality can inform the clinician about resources available for support (Osafo et al., 2021; CDC, 2022).

Another strengths-focused instrument was also not associated with suicide risk in the current study. However, this measure may still aid in assessing the broader clinical picture and can guide planning treatment. Evaluating patient's self-assessment of their ability to cope with suicidal thoughts in the future can provide valuable information to the clinician (Czyz et al., 2014, 2016).

Our study points to the value and validity of suicide screening in healthcare settings; however, assessment alone is not sufficient. Identification of high-risk patients must be paired with effective linkage to high-quality, evidence-based treatment. There have been slow but steady efforts to build the mental health system in Tanzania, which allows for the referral of people experiencing suicidal thinking to standard care. However, there is an immediate need to develop suicide prevention interventions that can utilize currently available resources efficiently while rapidly developing new services. This may also include enhanced mental health education in nursing and allied health professions, task sharing of treatment with non-specialists, and utilizing technological advancements such as telehealth to increase treatment capacity (Knettel et al., 2023a). It is important, however, to ensure that new services are developed while considering patients' safety, the evidence base of the treatments provided, and culturally informed development or adaptation of treatment approaches (Perera et al., 2020; Spanhel et al., 2021).

The study findings should be interpreted in light of the following limitations. Participants in this study were adults recruited from two HIV clinics in an urban setting with a relatively small sample size, and therefore, the results may not be generalizable to other settings, populations, or youth under the age of 18. Future studies may seek to identify whether the patterns of suicide risk we observed among PLWH are similar for people with other health conditions. The intention of this work was to assess the value of brief screening by healthcare workers other than psychologists and psychiatrists; however, it is important to note that these mental health professionals should still be engaged in the oversight of screening programs and in intervening when patients are identified with active risk of suicide. Effective task shifting generally involves brief treatment and assessment with the option to refer for higher levels of care when appropriate.

## Conclusions

Our findings showed that the C-SSRS Screen Version, a 6-item measure of suicidal intensity, was a strong predictor of active plan or intent to attempt suicide and a feasible strategy for suicide risk screening in a setting where a full clinical interview is rarely feasible. Items from the BRFL measure that assessed fear of death and concerns about social reactions to suicide also significantly predicted suicide risk. These brief measures are appropriate for administration by nonspecialists and show promise for identifying the risk of suicide in settings with few mental health providers. Implementing routine suicide risk screening in HIV care and other high-risk settings is a low-impact strategy to rapidly improve the assessment of suicide, enhance linkage to mental health care, and save lives.

**Open peer review.** To view the open peer review materials for this article, please visit http://doi.org/10.1017/gmh.2023.59.

**Supplementary material.** The supplementary material for this article can be found at http://doi.org/10.1017/gmh.2023.59.

**Data availability statement.** The de-identified data for this manuscript may be made available upon reasonable request to the authors with the completion of an appropriate data transfer agreement.

**Acknowledgements.** The authors would like to thank the staff at the study clinics for their support throughout the study period. The authors are also grateful to the participants in this study for sharing their experiences.

**Author contribution.** B.A.K., M.V.R., B.T.M., and D.B.G. conceived the study. I.A., L.J., C.S.T., and V.K. collected data. L.M., K.M., I.A., E.K., M.V.R., J.R.N.V., C.A.S., and E.F.M. contributed to the study design. L.M. analyzed the data, and B.A.K. and W.P. supervised the study. L.M. and K.M. wrote the first draft of the manuscript. B.A.K. and W.P. provided substantial editing and review. All authors read and revised or approved the final manuscript.

**Financial support.** Brandon Knettel is supported by a Career Development Award from the NIH National Institute of Mental Health (K08 MH124459).

The authors also acknowledge the support received from the grant, "Socio-behavioral Sciences Research to Improve Care for HIV Infection in Tanzania" (D43 TW009595) and the Duke Center for AIDS Research, an NIH-funded program (P30 AI064518).

**Competing interest.** The authors declare none.

**Ethics standard.** Study procedures were approved by the Tanzanian National Institute for Medical Research (R.8c/Vol.I/2122) and the ethical review boards of Kilimanjaro Christian Medical Centre (No. 2523) and Duke University Health System (Pro00107424).

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
