## [Reviewer Report]

Brandon A. Knettel, Ph.D.

Email: Brandon.Knettel@duke.edu

Prof. Judy Bass and Dr. Dixon Chibanda, Editors-in-Chief

Global Mental Health

May 4, 2023

Dear Dr. Bass, Dr. Chibanda, and Editorial Team:

Please see our enclosed manuscript, “Validation of a culturally sensitive instrument to assess suicide risk, reasons for living, and self-efficacy to avoid suicide with a sample of Tanzanian adults living with HIV,” which we are submitting for your consideration. Suicide is a leading cause of death among people living with HIV (PLWH) worldwide, yet few validated measures exist to assess suicide risk, particularly in low- and middle-income countries such as Tanzania. We adapted and conducted a thorough validation of items from three instruments to assess their utility for measuring suicide risk in this population. Through this process, we identified two subscales that can be used to conduct brief, valid, and reliable suicide risk screening among PLWH in Tanzania.

We believe that this manuscript is a strong fit for Global Mental Health and will be of interest to your readers. We present rigorous adaptation and validation procedures, and the resulting instrument shows excellent potential to enhance suicide risk screening for PLWH in low-resource settings.

The manuscript has not been previously published, nor has it been submitted for publication elsewhere. We declare that we have no conflicts of interest in relation to this research and all sources of support have been appropriately disclosed. The authors are the sole owners of the original data, which may be made available upon request. Institutional and national ethical approval were obtained prior to this study and all participants provided their written informed consent prior to enrollment. All listed authors contributed substantially to the study and manuscript according to ICMJE criteria.

Thank you for your consideration. We look forward to your review and to the possibility of having our work published in Global Mental Health.

Sincerely,

Brandon A. Knettel, Ph.D.

Assistant Professor and Licensed Psychologist

Associate Director, Duke Global Mental Health Program

Duke University School of Nursing and Duke Global Health Institute

---

## [Reviewer Report]

TITLE

Validation Of a Culturally Sensitive Instrument To Assess Suicide Risk, Reasons For Living, And Self-Efficacy To Avoid Suicide With a Sample Of Tanzanian Adults Living With HIV.

COMMENTS

General Comments

This study is useful and important. Africa needs more culturally validated mental screening instruments including assessing suicide risk. However, the manuscript was written in a haphazard manner. A scientific paper describing a research study must follow a standard script format: Introduction/Background, Objectives, Methods (and analysis), Results, Conclusions and Recommendations. Under each of these subheadings are specific items which must be included. This paper did not follow this time-honored format, left many gaps and made one wonder about the methodological correctness.

Specific Comments

Title: The title is too long. It should be less than 20 words. It seemed the authors wanted to include in it all the instruments used in the study. This is not necessary. The gist of this study was to validate the Swahili Translated Columbia Suicide Severity Rating Scale (C-SSRS) to assess suicide risk in PLWH in Tanzania. Example: Validation Of a Swahili Translated Culturally Sensitive Instrument To Assess Suicide Risk Among HIV-Positive Adults In Tanzania.

Abstract: This should be re-written in a structured way with subheadings of: Introduction, Objectives, Methods, Results, Conclusions.

Key words should be: HIV, Suicide Screening, Tanzania

Background: The background is not the place to discuss the study instruments. This goes into the Methods section. So lines 97 to 113 should be deleted. The last paragraph of the background should clearly spell out the problem which the study wants to address. Following this, it should clearly state the aim of the study as its stated objectives (which the authors did not clearly state here), followed by the hypothesis and what the findings will hopefully tell the reader.For example: “This study aimed to validate the Swahili Translated Columbia Suicide Severity Rating Scale (C-SSRS) to assess suicide risk in PLWH in Tanzania; assess their reasons for living and their self-efficacy to avoid suicidal actions. We hypothesized that……”

Methods: The Methods section should begin by describing the study design, then describe the study site (where, exactly, were the two HIV clinics located in Moshi?) followed by a description of the study participants with inclusion and exclusion criteria. The sampling frame should then follow, including sample size and how recruitment was done to avoid bias. However, the authors gave this statement at the very beginning: During routine HIV clinic appointments, professional nurses administered a single yes/no screening item derived from the C-SSRS Screen Version (Posner et al., 2011). The question was: “In the last month, have you had any actual thoughts of killing yourself?” Patients who responded yes to the screening item were informed of the research and, if interested, were referred to meet with a study Research Assistant; where then the participants were consented. This is procedurally troublesome. First, the question is from the study instrument; secondly, it is being asked before the study starts; and thirdly before the participant has been consented !!! All this points out that what was written in this paper was not described in a scientifically cohesive manner or that the methodological procedures were wrongly carried out . Indeed, the study procedures were difficult to follow i.e. what followed what and how? Also, the authors needed to describe the study instruments in detail e.g. what tool was used to collect the socio-demographic data? The C-SSRS was somehow described but not the other instruments i.e. the Brief Reasons for Living Inventory and the Self-Efficacy to Avoid Suicidal Action scale. Moreover, the descriptions of these instruments including their administration should be in the Methods Section, not in the Background Section.

The ethical protocol was not described in detail. How was consent obtained? How about participant privacy and data confidentiality and protection? How were actively suicidal participants handled? These present a psychiatric emergency. Mere simple counselling is not enough!! The RA and their roles were well described but not the other aspects of quality assurance.

This was a tool-validation study. What was the Gold Standard with which to compare it so as to draw ROCs? The description of the statistical analytic method used was difficult to follow and not reader friendly especially with no ROCs being presented to graphically show the AUC and cut off points. EFA was also difficult to follow. All this section needs to be reviewed by a statistician.

Results:These were presented in a rather confusing way. First, the authors needed to first present and describe the participants’ socio-demograhic characteristics. These would then be followed by the results addressing the research objectives: validation of the Swahili translated C-SSRS, including showing the graphs of ROCs demonstrating the AUC and the suicide risk predictions by the four subscales (cut off points), the SEASA & BRFL and finally the Exploratory factor analyses (EFA) showing the correlation with suicide risk based on Cronbach’s alpha.

Discussion: This discussion did not address the gist of this study which was to validate a Swahili translated C-SSRS. Instead it discussed generalities with statements like “Our study points to the value and validity of suicide screening in healthcare settings however, assessment alone is not sufficient. Identification of high-risk patients must be paired with effective linkage to high-quality, evidence-based treatment. There have been slow but steady efforts to build the mental health system in Tanzania….”; Such statements are not from this study. The authors need to stick to their objectives, findings and then compare and contrast with other studies validating suicide screening instruments and their utility in specific populations.This also goes to the conclusions and recommendations.

References: These are fair but there needed to be more references aligned to validating of suicide risk instruments.

---

## [Reviewer Report]

This is a well written and timely manuscript that details the development of validated Swahili translations of the Columbia Suicide Severity Rating Scale (C-SSRS) Screen Version and two accompanying scales assessing self-efficacy to avoid suicidal action and reasons for living among people living with HIV in Tanzania.

It would be helpful if the authors added additional context regarding their choice to add “the Brief Reasons for Living Inventory and the Self-Efficacy to Avoid Suicidal Action scale to the C-SSRS screener”. The authors state, “to capture a broader view of suicide risk”. Additional description of past work with the Brief Reasons and the Self-Efficacy measures in SSA or elsewhere would be helpful to the reader. Have other studies combined these measures with the CSSR?

Confusing or contradiction regarding reasons for living findings:

It seems contradictory that the authors write in the discussion “In our analysis, participants with low endorsement of fear of death and low social concern as reasons for living were more likely to have an active plan or intent to act on suicidal thoughts.”

Then in the next paragraph:

“The two additional subscales of Self-efficacy to Avoid Suicidal Action and Family and 316 Spirituality as a reason for living were not associated with suicide risk.”

And in the conclusion: “Items from the reasons for living measure that assessed fear of death and concerns about social reactions to suicide also significantly predicted suicide risk.”

Given the time and resource constraints, do the authors think that the additional measures were beneficial or might they recommend just the CSSRS? I do see the current paragraph re the benefits of additional clinical context.

Minor:

Colombia Suicide Severity 175 Rating Scale (C-SSRS)

Spelling: “Columbia”

Line 328 missing word “there” ? “However, is an immediate need to develop”

BRFL and reasons for living are used interchangeable, is the acronym necessary? If so perhaps use throughout, if not then just use reasons for living to be consistent.

---

## [Reviewer Report]

Brandon A. Knettel, Ph.D.

Email: Brandon.Knettel@duke.edu

Prof. Judy Bass and Dr. Dixon Chibanda, Editors-in-Chief

Global Mental Health

REF: GMH-23-0094

September 5, 2023

Dear Dr. Bass, Dr. Chibanda, and Editorial Team:

My co-authors and I appreciate your review of our manuscript, and the invitation for us to revise and resubmit this work. Below you will find our responses to the reviewer’s comments, with page numbers referring to the revised manuscript. Revisions in the manuscript were made using Track Changes and a clean copy of the revised manuscript is also provided.

We sincerely appreciate the thoughtfulness of the reviewers and your recognition of the importance of this topic. The questions and suggestions have guided our conceptualization and presentation of our findings and have greatly enhanced the quality and clarity of the manuscript. 

We look forward to your review and to the possibility of having our work published in Global Mental Health.

Sincerely,

Brandon A. Knettel, Ph.D.

Duke University School of Nursing and Duke Global Health Institute